

# Comparison of perceived masticatory ability in completely edentulous patients treated with thermoplastic complete denture versus single implant-retained mandibular overdenture: a single-center prospective observational study

Mostafa I. Fayad[1,2], Ihab I. Mahmoud[2], Ahmed Atef Aly Shon[2,3], Mohamed Omar Elboraey[4,5], Ramy M. Bakr[6] and Rania Moussa[1]

[1] Substitutive Dental Sciences Department, College of Dentistry, Taibah University, Madina, Saudi Arabia
[2] Removable Peosthodontic Department, Faculty of Dental Medicine, Al-Azhar University, Cairo, Egypt
[3] Prosthodontic Department, Mouwsat Hospital, Medina, Saudi Arabia
[4] Oral Medicine, Periodontology, Oral Diagnosis and Radiology Department, Faculty of Dentistry, Tanta University, Tanta, Egypt
[5] Periodontology and Preventive Dental Science Department, College of Dentistry, Taibah University, Madina, Saudi Arabia
[6] Removable Prosthodontic Department, College of Dentistry, Future University, Cairo, Egypt

Corresponding author
Mostafa I. Fayad,
mifayad@taibahu.edu.sa

## ABSTRACT

**Background**. This study aimed to compare the perceived masticatory ability (PrMA) in completely edentulous patients (EDPs) with thermoplastic conventional complete dentures (CDs) versus single implant-retained mandibular overdentures.

**Methods**. The current study was conducted in the outpatient Prosthodontic Clinic, Faculty of Dental Medicine, Al-Azhar University, Cairo, Egypt. PrMA was evaluated in 45 completely edentulous patients (46% males, mean age 50.4 ± 4.7 years). Each patient received a thermoplastic PMMA complete denture (Polyan IC TM Bredent GmbH & Co.KG, Germany). The PrMA was evaluated at one-month and six-month intervals of denture use. An immediate loading single implant was placed into the mid-symphyseal for each patient, and the denture was adjusted. Subsequently, the PrMA was reevaluated after one month and six months. The data were collected and statistically analyzed using the SPSS@V25 to assess the changes in PrMA.

**Results**. The PrMA demonstrated improvement after six months of thermoplastic conventional denture use. However, this improvement was not statistically significant ($p = 0.405$). In addition, the PrMA showed a substantial increase following a single implant placement at one and six months ($p < 0.001$) of the overdenture use compared to the conventional denture. The PrMA insignificantly improved ($p = 0.397$) after six months of the single implant retained overdenture use.

**Discussion**. The study's findings indicate that using immediate loading single implant-retained mandibular overdentures significantly improved PrMA in completely edentulous patients.

## INTRODUCTION

The main objective of prosthodontic rehabilitation is to restore and maintain oral function, especially the effectiveness of masticatory function (*Liang et al., 2015*). Edentulous patients wearing conventional dentures often experience a significant decrease in their ability to chew well, which impacts their quality of life. Furthermore, the complex neuromuscular skills required to overcome the limitations of dentures decline with age (*Goiato et al., 2008*).

Dentures with inadequate masticatory efficiency prevent wearers from effectively consuming high-fiber diets. Consequently, dentures must facilitate efficient chewing (*van der Bilt, 2011*; *van der Bilt & Fontijn-Tekamp, 2004*).

Denture wearers may experience limited chewing force due to the discomfort and pain that occurs when one or both dentures lose their retention or even fear of pain (*Goiato et al., 2010*).

The perception of chewing ability among individuals is strongly correlated with their oral health-related quality of life (OHRQoL). Patients with higher oral health impact profile (OHIP) scores are more prone to having chewing problems and perceived difficulty (*Khalifa et al., 2013*). Improving masticatory performance in patients with CDs also benefits their overall well-being (*Elmoula, Khalifa & Alhajj, 2018*).

The masticatory function can be addressed from two perspectives: firstly, as the ability to objectively break down solid food, and secondly, as an individual's personal response when queried about their food-chewing experiences. Masticatory performance, the objective measure of masticatory function, is often assessed by determining an individual's ability to pulverize or grind a designated food item within a predetermined number of chewing cycles. The studies examined the self-assessed masticatory function of the participants (defined as masticatory ability) through oral function interviews (*Feizi et al., 2016*; *van der Bilt, 2011*).

Several objective techniques have been attempted to assess masticatory performance. However, they need specialized tools, materials, or intricate procedures. Experiments investigating masticatory performance have utilized natural foods, such as almonds, peanuts, and carrots, and synthetic materials as test substances. (*Cunha et al., 2013*; *Goiato et al., 2008*; *Liedberg & Owall, 1995*; *van der Bilt & Fontijn-Tekamp, 2004*).

Another commonly utilized approach to assess masticatory performance involves evaluating the capacity to blend and manipulate a meal bolus thoroughly. The masticatory performance has been assessed using two-colored chewing gum and paraffin wax as test meals (*Salleh et al., 2007*; *van der Bilt, 2011*).

Both subjective and objective methods can be effectively used in measuring masticatory performance. *Elmoula, Khalifa & Alhajj (2018)* found a correlation between the subjectively evaluated PrMA and the objectively assessed masticatory efficiency.

The masticatory function of the complete-denture wearers is relatively poor compared to that of healthy dentate subjects. Complete-denture wearers need up to seven times more chewing strokes than subjects with a complete natural dentition to reduce the food to half the original particle size (*Emami et al., 2013*; *Kumari et al., 2022*). Other studies have reported that patients rehabilitated with CDs demonstrated significantly lower masticatory functions (*Slagter et al., 1992*; *Wayler & Chauncey, 1983*)

When conventional denture therapy is inadequate, it is essential to examine treatment alternatives to improve the masticatory efficiency of complete denture wearers. An alternative method is the utilization of the injection-molded thermoplastic denture base (*Fayad, Mahmoud & Shon, 2023*).

The injection-molded PMMA has a micro-crystalline structure, which ultimately facilitates the process of finishing and polishing. The use of the injection molding method for fabricating dentures resulted in enhanced quality and durability. This was attributed to the higher microhardness and reduced surface roughness compared to conventional denture bases (*Moslehifard et al., 2022*).

Thermoplastic denture bases exhibit superior aesthetics and are more embraced by patients than conventional dentures. They can serve as a substitute for individuals who have allergic reactions to polymethyl methacrylate. Due to their low weight and pliable characteristics, they can be effectively used on individuals with skeletal protuberances. The material's flexibility provided a targeted stress reduction level, eliminating denture-related problems that were causing oral discomfort (*Singh et al., 2011*).

Another alternative to improve the masticatory functions for completely edentulous patients is implant placement to improve denture retention and stability, thereby improving masticatory performance (*Bae et al., 2015*; *Fayad et al., 2016*; *Mohamed, 2008*). Despite the growing use of osseointegrated implants in rehabilitation, conventional CDs remain the most common treatment method for completely edentulous patients, especially in underdeveloped countries (*Carlsson & Omar, 2010*).

Numerous clinical studies have shown that utilizing implant-supported or retained prostheses for rehabilitating the mandible in individuals without teeth has proven a highly effective and gratifying treatment (*Kourtis et al., 2018*). Nevertheless, there is an ongoing debate regarding the minimal number of implants required for this restoration. The single implant retained overdenture has become increasingly popular due to its simple technique (*Mahoorkar, Bhat & Kant, 2016*).

It has been hypothesized that placing a single implant in the middle of the symphyseal region can effectively support an overdenture with a high success rate based on Albrektsson's success criterion (*Albrektsson & Wennerberg, 2019*; *Gjelvold et al., 2020*). This treatment approach can also be a cost-effective therapeutic alternative to the traditional complete denture (*Krennmair & Ulm, 2001*; *Passia & Kern, 2023*).

In a study conducted by *Liu et al. (2013)* on the implant number required to retain mandibular implant-retained overdenture, it was found that a single implant is sufficient to support and distribute the load effectively to the mandibular bone in implant-retained overdentures.

To our knowledge, no studies have assessed the perceived masticatory ability (PrMA) among completely edentulous patients rehabilitated with a thermoplastic acrylic denture before and after placing a single implant. This study aimed to determine the changes in the PrMA after single implant placement in completely edentulous patients. The null hypothesis was that the placement of a single implant to retain a complete mandibular thermoplastic denture would not affect the PrMA.

## MATERIALS AND METHODS

This study was conducted at the Faculty of Dental Medicine, Al-Azhar University, Cairo, Egypt, using a prospective study design. The study was conducted over a period of 18 months, spanning from April 2022 to August 2023. The ethics committee at Al-Azhar University has approved the study protocol (Ethical Application Ref: AUAREC20220004-12). Before their enrollment in the study, all participants received a detailed explanation of the methodology. Subsequently, written consent was obtained from all participants.

### Patients' selection

All patients included in the study were free of any psychiatric problems or movement disorders. Patients who have previously had temporomandibular problems, including Myofacial Pain Dysfunction Syndrome (MPDS), trismus, trauma, TMJ dislocation, and ankylosis, were not included in the study. Furthermore, those with compromised oral diseases, local lesions, and resorbed or flabby ridges were excluded.

Patients with oral diseases that may compromise the masticatory function were excluded from the study. Due to the detrimental impact of xerostomia on quality of life and its correlation with decreased masticatory function (*Moriya et al., 2012*), patients with xerostomia were also excluded from the study.

Prior researches have determined that a sample size of 40 cases is adequate to conduct the study with a statistical power of 0.80, a confidence interval of 0.95, and an alpha level of 0.05 (*Albert, Buschang & Throckmorton, 2003*; *Goiato et al., 2010*; *Mohamed, 2008*; *Tatematsu et al., 2004*). Consequently, a higher sample size calculation was determined ($n = 50$) to compensate for the possibility of edentulous participants' withdrawal due to illness, death, or challenges with the research protocol.

A total of 50 completely edentulous patients were chosen randomly. Five patients withdrew from the study, so only 45 patients were evaluated. The group consisted of 21 males and 24 females, with an age range of 44–59 years (mean age 50.4 ± 4.77 years).

All patients received a new thermoplastic PMMA conventional complete denture (Polyan IC TM Bredent GmbH & Co.KG, Senden, Germany), with even occlusion and discomfort-free. The new complete dentures (CDs) were delivered and evaluated over a period of one month to ensure there was no reported pain or discomfort (*Mathew et al., 2024*; *Rocha et al., 2023*).

### First stage-measurement of the PrMA

The subjective approach to evaluating masticatory ability was assessing the PrMA. The measurement was conducted using a perceived difficulty of chewing (PDC) index score

devised by *Khalifa et al. (2013)*. Participants were asked to report the level of difficulty they experienced while chewing fifteen commonly consumed hard and soft foods. The scoring of this index was determined for each food type based on (PDC) scale, with a range of scores from 0 (indicating very easy chewing) to 5 (indicating very difficult chewing that is actively avoided). A total score of zero indicates very easy chewing and satisfactory conditions, whereas a total score 75 signifies adverse conditions and the most difficult chewing.

The PrMA was measured for each patient after one month of conventional thermoplastics denture placement. The second measurement was conducted six months following the conventional thermoplastic denture placement, as recommended by Goiato (*Goiato et al., 2010*; *Goiato et al., 2008*). It was proposed that a minimum of five months was required to assess patient adaptability and functional capacity with new CDs adequately.

### Mid-symphyseal single implant placement

Cone-beam computed tomography (CBCT) scans of the mandible were performed for each patient using the Kodak 9500 cone-beam 3D System scanner manufactured by Carestream Dental/Kodak in the United States. For each patient, a mid-symphyseal dental implant was placed (Dentis; Dalseo-gu, Daegu, Korea). The mandibular denture was prepared for insertion following a two-day implant placement period. The locator attachment (Dentis; Dalseo-gu, Daegu, Korea) was affixed to the fixture and secured with a screwdriver.

The resilient cap was placed over the male part of the attachment and then transferred to the base of the denture using a marker on the cap. Subsequently, the lower denture was inserted in the patient's mouth, marking the corresponding cap area on the fitting surface of the denture. The resilient cap (female part) housing was formed on the fitting surface of the denture in the designated area using a round bur rotating at a low speed.

The denture was examined in the patient's mouth to ensure the absence of interference. Auto-polymerizing acrylic resin was placed in the space created in the denture base. A small amount of resin was injected intraorally into the dry metallic cap.

The denture was placed in the patient's mouth, and the patient was advised to close their mouth, causing the metal cap to be fitted into the base of the denture. After the acrylic resin had solidified, the denture was removed from the mouth and examined, and any surplus material was eliminated using an appropriate bur.

### Second stage-measurement of the PrMA

The PrMA was measured for each patient after one month of single implant-retained mandibular overdenture placement, and the final measurement was conducted after six months.

### Statistical analysis

Data were collected, and the statistical analysis was conducted using IBM SPSS Statistics V25 software (Armonk, NY: IBM Corp). The level of statistical significance was set at 0.05 for all tests. The normality of continuous data was assessed using the Shapiro–Wilk test. Quantitative data were expressed as range (minimum and maximum), mean, standard deviation, and median. Descriptive statistics of mean and standard deviation were reported.

**Table 1 Gender frequency.**

| | | Frequency | Percent | Valid percent | Cumulative percent |
|---|---|---|---|---|---|
| | Male | 21 | 46.7 | 46.7 | 46.7 |
| Valid | Female | 24 | 53.3 | 53.3 | 100.0 |
| | Total | 45 | 100.0 | 100.0 | |

The Mann–Whitney test compared two groups with non-normally distributed quantitative variables.

In contrast, the Kruskal–Wallis test compared different groups with non-normally distributed quantitative variables. The Friedman tests were employed to compare quantitative variables that do not follow a normal distribution across more than two periods or stages. The *post-hoc* paired comparison was conducted utilizing the Wilcoxon signed-rank test. The statistical significance of the obtained results was judged at the 5% level.

## RESULTS

The PrMA was evaluated among completely edentulous patients using a thermoplastic PMMA denture base at one month and six months of complete denture placement. After placing a single implant-retained mandibular overdenture, the PrMA was reevaluated at one month and six months.

The sample included 50 completely edentulous patients who were randomly selected. A total of 45 patients were assessed, with one being discharged due to medical issues and four opting not to continue with the study. The patients comprised 21 male and 24 female patients (Table 1). The mean age of the selected patients was 50.46 years, ranging from 44 to 59 years.

Table 2 shows the mean and standard deviation of the PrMA measurements at different intervals. The PrMA for each participant was obtained by collecting each food PrMA score (from 0 to 5). The mean value for PrMA one month and six months following the placement of the new denture was $37.8 \pm 10.5$ and $36.3 \pm 10.3$, respectively. The mean value for PrMA after the single implant placement at one month and six months was $28.6 \pm 8.4$ and $26.9 \pm 8.5$, respectively (Table 2).

The results of the Kolmogorov–Smirnov and Shapiro–Wilk tests which were used to assess the normality of the data (*Kim, 2012*; *Kim, 2013*), showed that the data were not normally distributed, as illustrated in Table 3.

The nonparametric Friedman test was used for within-subject design due to the non-normal distribution of the data. The *post-hoc* paired comparison was done using the Wilcoxon signed rank test [30, 31]. The Friedman test (Table 4) showed a statistical significance difference between different measurements of PrMA at various intervals.

The multiple comparisons between different mean measurements of PrMA at various intervals (Table 5) showed no statistical difference in PrMA recorded after one month of denture insertion and PrMA recorded after six months of denture insertion ($P > 0.05$).

**Table 2** The mean and standard deviation of the perceived masticatory ability measurements at different intervals.

| Evaluation intervals | Gender | Mean | SD | Std. Error | Minimum | Maximum |
|---|---|---|---|---|---|---|
| Con1 | Male | 38.90 | 10.95 | 2.39 | 15.00 | 55.00 |
| | Female | 37.00 | 10.40 | 2.12 | 15.00 | 50.00 |
| | Total | 37.88 | 10.58 | 1.57 | 15.00 | 55.00 |
| Con6 | Male | 37.19 | 10.26 | 2.23 | 15.00 | 53.00 |
| | Female | 35.62 | 10.66 | 2.17 | 15.00 | 50.00 |
| | Total | 36.35 | 10.39 | 1.54 | 15.00 | 53.00 |
| impl1 | Male | 29.33 | 8.39 | 1.83 | 15.00 | 50.00 |
| | Female | 28.08 | 8.58 | 1.75 | 10.00 | 39.00 |
| | Total | 28.66 | 8.42 | 1.25 | 10.00 | 50.00 |
| imp6 | Male | 27.52 | 9.00 | 1.96 | 13.00 | 43.00 |
| | Female | 26.45 | 8.21 | 1.67 | 9.00 | 38.00 |
| | Total | 26.95 | 8.50 | 1.26 | 9.00 | 43.00 |

**Notes.**

Con1, PrMA recorded after one month of thermoplastic complete denture placement; Con6, PrMA recorded after six months of thermoplastic complete denture placement; Impl1, PrMA recorded after one month of single implant-retained mandibular overdenture placement; Imp6, PrMA recorded after six months of single implant-retained mandibular overdenture placement.

**Table 3** Tests of normality.

| | Kolmogorov–Smirnov[a] | | | Shapiro–Wilk | | |
|---|---|---|---|---|---|---|
| | Statistic | df | Sig. | Statistic | df | Sig. |
| Mast | .092 | 178 | .001 | .974 | 178 | .002 |

**Notes.**

[a] Lilliefors significance correction.

**Table 4** Friedman test.

| Ranks | | Test statistics | | | | |
|---|---|---|---|---|---|---|
| | Mean Rank | N | Chi-Square | df | Asymp. Sig. | |
| Con1 | 3.53 | | | | | |
| Con6 | 3.23 | 45 | 96.60 | 3.00 | 0.000 | |
| impl1 | 1.72 | | | | | |
| imp6 | 1.51 | | | | | |

**Notes.**

Con1, PrMA recorded after one month of thermoplastic complete denture placement; Con6, PrMA y recorded after six months of thermoplastic complete denture placement; Impl1, PrMA recorded after one month of single implant-retained mandibular overdenture placement; Imp6, PrMA recorded after six months of single implant-retained mandibular overdenture placement.

Mid-symphyseal single Implant placement resulted in a substantial increase in the PrMA. In addition, there was a highly statistically significant difference between the PrMA recorded before and after single implant placement ($P < 0.05$).

The study sample was subdivided into three subgroups based on age range: (1) <47 ($n = 13$), (2) from 47–52 ($n = 14$), and (3) >52 ($n = 18$). The Mann–Whitney test (Table 6)

**Table 5 Perceived masticatory ability means comparison at different intervals.**

| (I) test | (J) test | Mean Difference (I-J) | Std. error | Sig. | 95% Confidence Interval | |
|---|---|---|---|---|---|---|
| | | | | | Lower Bound | Upper Bound |
| Con1 | Con6 | −1.702 | 2.039 | .405 | −2.32 | −5.72 |
| | Impl1 | −9.222* | 2.015 | .000* | −5.24 | −13.20 |
| | Impl6 | −10933* | 2.015 | .000* | −6.95 | −14.91 |
| Con6 | Con1 | −1.702 | 2.039 | .405 | −5.72 | −2.32 |
| | Impl1 | −7.519* | 2.039 | .000* | −3.49 | −11.54 |
| | Impl6 | −9.230* | 2.039 | .000* | −5.20 | −13.25 |
| Impl1 | Con1 | −9.222* | 2.015 | .000* | −13.20 | −5.24 |
| | Con6 | −7.519* | 2.039 | .000* | −11.54 | −3.49 |
| | Impl6 | −1.711 | 2.015 | .397 | −2.26 | −5.68 |
| Impl6 | Con1 | −10933* | 2.015 | .000* | −14.91 | −6.95 |
| | Con6 | −9.230* | 2.039 | .000* | −13.25 | −5.20 |
| | IImpl1 | −1.711 | 2.015 | .397 | −5.68 | −2.26 |

**Notes.**
*The mean difference is significant at the 0.05 level.
Con1, PrMA recorded after one month of thermoplastic complete denture placement; Con6, PrMA recorded after six months of thermoplastic complete denture placement; Impl1, PrMA recorded after one month of single implant-retained mandibular overdenture placement; Imp6, PrMA recorded after six months of single implant-retained mandibular overdenture placement.

**Table 6 Relation between gender and PrMA.**

| | Sex | | U | p |
|---|---|---|---|---|
| | Male (n = 21) | Female (n = 24) | | |
| **Con1** | | | | |
| Mean ± SD. | 38.9 ± 11 | 37 ± 10.4 | 238.0 | 0.747 |
| Median (Min.–Max.) | 40 (15–55) | 40 (15–50) | | |
| **Con6** | | | | |
| Mean ± SD. | 37.2 ± 10.3 | 35.6 ± 10.7 | 233.0 | 0.658 |
| Median (Min.–Max.) | 40 (15–53) | 40 (15–50) | | |
| **Impl1** | | | | |
| Mean ± SD. | 29.3 ± 8.4 | 28.1 ± 8.6 | 249.50 | 0.954 |
| Median (Min.–Max.) | 30 (15–50) | 30 (10–39) | | |
| **Imp6** | | | | |
| Mean ± SD. | 27.5 ± 9 | 26.5 ± 8.2 | 238.0 | 0.746 |
| Median (Min.–Max.) | 28 (13–43) | 28 (9–38) | | |

**Notes.**
SD, Standard deviation; U, Mann Whitney test; p, p value for comparing between male and female.

was used to test the effect of gender on the PrMA at different intervals. The results showed no statistically significant effect of gender on the PrMA at various intervals. The Kruskal–Wallis test (Table 7) was used to test the impact of different age groups on the

**Table 7   Relation between different age group and PrMA.**

| | Age | | | H | p |
|---|---|---|---|---|---|
| | Less than 47 (n = 13) | 47 to 52 (n = 14) | More than 52 (n = 18) | | |
| **Con1** | | | | | |
| Mean ± SD. | 36.9 ± 10.7 | 34.1 ± 11.9 | 41.6 ± 8.6 | 4.377 | 0.112 |
| Median (Min.–Max.) | 40 (15–55) | 39 (15–54) | 45 (20–55) | | |
| **Con6** | | | | | |
| Mean ± SD. | 35.6 ± 11.1 | 32.1 ± 11.9 | 40.2 ± 7.4 | 4.396 | 0.111 |
| Median (Min.–Max.) | 35 (15–53) | 37.5 (15–50) | 40 (15–50) | | |
| **Impl1** | | | | | |
| Mean ± SD. | 27.4 ± 7.3 | 26.4 ± 9.4 | 31.3 ± 8.1 | 2.110 | 0.348 |
| Median (Min.–Max.) | 30 (15–40) | 30 (12–35) | 30 (10–50) | | |
| **Imp6** | | | | | |
| Mean ± SD. | 25.6 ± 9.3 | 24.6 ± 8.9 | 29.7 ± 7.3 | 3.623 | 0.163 |
| Median (Min.–Max.) | 23 (13–43) | 28 (11–38) | 29 (9–43) | | |

**Notes.**
SD, Standard deviation; H, H for Kruskal–Wallis test; $p$, $p$ value for comparison between the studied categories.

PrMA at different intervals. The findings indicated that gender did not significantly affect the PrMA at various time intervals.

## DISCUSSION

The current study's results reported that the placement of a single mid-symphyseal implant significantly affected the PrMA of the study groups. Therefore, the null hypothesis has been rejected and there was strong evidence of a difference between the groups.

In this study, the masticatory ability was assessed using questionnaires. However, this method needs to be objective for repeatability. Consequently, it is more reasonable to evaluate the masticatory 304 function using a combination of questionnaires and clinical assessments. Previous studies indicated that subjective evaluation of self-perceived chewing ability was as valid as objectively assessed masticatory efficiency. Both methods have proven equally effective in clinical practice (*Limpuangthip, Somkotra & Arksornnukit, 2021*).

In the case of a completely edentulous wearer, the subjective criteria may be more critical than the chewing tests. Therefore, questionnaires are regarded as a valuable tool (*Boretti, Bickel & Geering, 1995*). In addition, in complete denture wearers, the subjective criteria may be additionally explanatory as the complete denture quality has been significantly related to patient satisfaction and perceived chewing ability (*Yamaga, Sato & Minakuchi, 2013*).

The perceived masticatory index for each participant was determined using natural test foods due to their regular consumption in daily life and familiarity with patients (*Mathew et al., 2024*).

The thermoplastic denture base material selected in this study was based on its utilization of the injection molding technique, which allows for a controlled polymerization process. The flask design facilitates a constant flow of material through the sprue channel, thereby

compensating for polymerization shrinkage and yielding superior dimensional accuracy compared to compression molding (*Khan et al., 2022*). It also shows significantly better flexural strength and higher flexural modulus, resulting in minimal deformation before fracture (*Patankar et al., 2022*).

Multiple clinical studies have verified that the adaptation period for both new CDs and new mini-implant overdentures opposing maxillary CDs is typically one month. The PrMA was assessed one month after the denture placement, as documented by previous studies (*Hayakawa et al., 2000*; *Poljak-Guberina et al., 2022*; *Topic et al., 2022*). The second measurement was conducted after six months, as recommended by previous studies (*Goiato et al., 2008*; *Goiato et al., 2010*), which suggested that more than five months were needed to evaluate patient adaptation and functional capacity with new CDs.

The immediate loading implant procedure has demonstrated reliability and effectiveness in various clinical contexts. It reduces the treatment time by the possibility of immediate implant functionality by positioning within 48–72 h after fixture placement (*Mangano et al., 2017*; *Raes et al., 2018*). Loading single implants has proven its efficacy and reliability as a treatment approach (*Raes et al., 2018*).

Single implant placement has been suggested to address some of the forthcoming limitations of using two or more implants. The two-implant overdenture has demonstrated efficacy and was proposed as the minimum standard of treatment that should be offered to completely edentulous mandible patients. However, the current increase in dental initial and ongoing maintenance makes the two-implant overdenture inaccessible to a significant number of financially disadvantaged elderly individuals (*Mathew et al., 2024*).

Studies anticipated the chair side time and the cost of fabricating the two-implant overdenture to be 1.75 times more than single-implant overdenture. However, both demonstrated adequate clinical efficacy and patient satisfaction (*Mahoorkar, Bhat & Kant, 2016*). The novelty of the current study is that the treatment provided to the study group comprised the advantages of the resilient thermoplastic resin and its cushioning effect. This resulted in enhanced support and retention offered by the dental implant.

After six months of denture placement, the results revealed an improvement in the masticatory function with a conventional complete thermoplastic denture. Furthermore, regarding single implant placement, an improvement was observed after six months of single implant-retained mandibular overdenture. However, there was no statistically significant difference (Table 5). The improvement may be attributed to increasing adaptation and subsequent denture stability after six months of use.

This result contradicts the findings of *Hazari et al. (2015)*. They found a statistically significant difference after six months, which may be attributed to their study's different assessments and thermoplastic materials. This improvement is highly substantial since complete thermoplastic dentures offer a more straightforward and cost-effective treatment alternative than other options, such as implant-supported dentures. Moreover, they substantially improve stability and retention for patients who struggle with adapting to conventional mandibular dentures. These results are consistent with the study conducted by *Berretin-Felix et al. (2008)*, who illustrated that the type of dental treatment used directly correlates with masticatory efficiency.

This study showed a significant difference in PrMA evaluated after six months of conventional denture placement compared to the perceived masticatory ability assessed after six months of single implant-retained mandibular overdenture. This finding corroborates *Rocha et al. (2023)*, who found that the treatment with mandibular overdentures supported by a single implant in the mandibular symphysis region improved masticatory efficiency over conventional CDs.

There was a significant difference between the evaluation of PrMA after six months of using conventional dentures and the assessment done one month after using a single implant retained mandibular overdenture (Table 3). This finding demonstrates the considerable enhancement following the placement of a single implant. Additionally, the masticatory function significantly improves after treating mandibular implant overdentures. Most studies on implant treatment and oral function showed a significant improvement of the objective masticatory performance in the mandibular overdenture (*Fontijn-Tekamp et al., 2004*).

The study of *Rocha et al. (2023)* evaluated the masticatory function objectively. It confirmed the importance of using a single implant to improve the masticatory function for completely edentulous patients.

The findings of this study indicated that gender had no impact on the PrMA (Table 6), which aligns with the results of *Elmoula, Khalifa & Alhajj (2018)*. In addition, the results showed that the various age groups within the study sample had an insignificant effect on the PrMA (Table 7). These results are inconsistent with those of *Hirai et al. (1994)*, who investigated the age-related changes in masticatory function in complete denture wearers. They found that both the masticatory performance and the chewing score decreased significantly due to aging. This finding may be attributed to the different age ranges of patients selected in this study.

This study's limitation is that it did not assess the impact of alveolar ridge height and denture retention on the results. Furthermore, it is essential to consider the correlation between self-assessed masticatory ability (SAMA) and psychological status. The findings of *Roohafza et al. (2016)* provide evidence that participants with a higher score of depression, anxiety, and stress experience decreased masticatory ability.

Therefore, future investigations should prioritize an integrated approach encompassing many aspects and incorporating dental care with other treatments, such as nutritional counseling, to improve eating habits and patients' quality of life.

It is also crucial to highlight the diagnostic aspect and preexisting preparation before denture fabrication. The human factors in planning and technical performance are decisive for rehabilitation success.

## CONCLUSION

The study demonstrated a significant improvement in PrMA in completely edentulous patients after rehabilitation with single implant-retained mandibular overdentures.

### Funding

The authors received no funding for this work.

### Competing Interests

The authors declare there are no competing interests.

### Author Contributions

- Mostafa I. Fayad conceived and designed the experiments, performed the experiments, analyzed the data, authored or reviewed drafts of the article, and approved the final draft.
- Ihab I. Mahmoud performed the experiments, prepared figures and/or tables, and approved the final draft.
- Ahmed Atef Aly Shon performed the experiments, prepared figures and/or tables, and approved the final draft.
- Mohamed Omar Elboraey performed the experiments, authored or reviewed drafts of the article, and approved the final draft.
- Ramy M. Bakr analyzed the data, prepared figures and/or tables, and approved the final draft.
- Rania Moussa conceived and designed the experiments, performed the experiments, analyzed the data, authored or reviewed drafts of the article, and approved the final draft.

### Human Ethics

The following information was supplied relating to ethical approvals (i.e., approving body and any reference numbers):

Faculty of Dental Medicine. Al-Azhar University granted Ethical approval to carry out the study within its facilities. (Ethical Application Ref: (AUAREC20220004-12)

### Data Availability

Raw data showing all perceived masticatory ability measurements at different intervals are available in the Supplemental Files.

### Supplemental Information

Supplemental information for this article can be found online at http://dx.doi.org/10.7717/peerj.17670#supplemental-information.

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
