# Peer review of "Comparison of perceived masticatory ability in completely edentulous patients treated with thermoplastic complete denture versus single implant-retained mandibular overdenture: a single-center prospective observational study"

_PeerJ, doi:10.7717/peerj.17670_

## Round 0.1 · original submission · Major Revisions

Please ensure that you take into consideration all the comments made by the reviewers and make the necessary changes accordingly.

·

Basic reporting

Thank you for handing in this great article. Regarding basic reporting, most of the PeerJ criteria are met: Literature references and sufficient context are provided. The article is structured well, figures and tables support the main messages of the article. The raw data is shared. The article is self-contained, with relevant results to hypotheses. I want to make the following additional comments:

1) I suggest mentioning the study type in the title and rewriting it slightly for better readability: “Comparison of perceived masticatory ability in completely edentulous patients treated with thermoplastic complete denture versus single implant-retained mandibular overdenture: A single-center prospective observational study”

2) Please revise the article for minor spelling errors that impact readability such as:
— Remove “average” from “average age range” in line 170.
— In lines 210 and 212 replace “implat” with “implant”.
— Recheck the following paragraph (lines 218-222) for readability and uppercase/lowercase writing: “Mann Whitney test was used to compare two groups for not normally distributed quantitative variables while Kruskal Wallis test was used to compare different groups for not normally distributed quantitative variables. while Friedman test For abnormally distributed quantitative variables, to compare between more than two periods or stages.”
— Please revise the sentence in line 226 “The est (pairwise comparison) is output for this purpose.”
— Please correct “The final measurment” in line 211.
— Insert the missing “o” letter in “thermplastic” in lines 185 and 187.
— Please recheck the address of the corresponding author: “Media”.

3) I suggest adding age and gender to the description of the study population in the abstract methods sections, such as: “45 completely edentulous patients (46% males, mean age 50.4 ± 4.77 years).”

4) I suggest replacing “which is a big concern for them” in line 64: “Edentulous individuals wearing conventional dentures often experience a significant decrease in their ability to chew well, which impacts their quality of life.”

5) In line 76 add the full description “Oral Health Impact Profile” to the acronym OHIP.

6) Please revise “E.G. written consent was obtained from all participants .”

7) Please revise “Patients' selection” as “Patient Selection” might be probably meant here.

Experimental design

The content of this article is within the aims and scope of the PeerJ journal. The research question is well-defined, relevant, and meaningful. It is stated how research fills an identified knowledge gap. The investigation has been performed according to technical and ethical standards. The authors describe the methods with sufficient detail.

Validity of the findings

The underlying data have been provided. The results can be reproduced based on the raw data. Conclusions are well stated, linked to the original research question and limited to supporting results.

Additional comments

No additional comments.

·

Basic reporting

The article needs to be reviewed by a fluent English speaker. I made some notes with some suggestions in the PDF. The article is well structured, good proposal, nice conclusions.

Experimental design

Research question well defined, relevant & meaningful.

Validity of the findings

Good results.

·

Basic reporting

1. The manuscript is well-written in professional and unambiguous English. However, there are a few instances where the language could be improved for better clarity and readability. For instance, on page 2, lines 66-68, page 3, lines 80-82, and page 4, lines 118-121, the language could be improved. As a solution, I suggest having the manuscript proofread by a fluent English speaker or a professional editing service.
2. The introduction provides some context, but it would be helpful to have a more comprehensive literature review. This would better highlight the existing knowledge gaps that your study aims to address, specifically on Page 2-3, lines 60-86. Additionally, it is recommended to expand on the importance of masticatory ability in completely edentulous patients. Furthermore, it would be beneficial to discuss the potential benefits of using thermoplastic dentures and single implant-retained mandibular overdentures.
3. The manuscript adheres to PeerJ standards, but ensure proper reference formatting and minimize self-citations (Page 10-12, lines 382-523).
4. The tables are well-described and relevant. It may be useful to present the results of normality tests (Kolmogorov-Smirnov and Shapiro-Wilk) in a table format to enhance readability (Page 6, lines 248-250).
5. Thank you for providing the raw data. Your effort is appreciated.

Experimental design

1. The study represents an original piece of primary research and is well within the scope of the journal.
The research question is well-defined, relevant, and meaningful. However, the introduction could be improved by providing more context on how this study fills an identified knowledge gap, even though the research question is well-defined, relevant, and meaningful. (Page 2-3, lines 60-86)
2. The investigation was rigorous, and the methods were described with sufficient detail to allow replication.

Validity of the findings

1. The study seems to include an appropriate sample size and number of participants.
2. The data appears to be robust, statistically sound, and controlled.
3. The conclusion should accurately reflect the study's findings and avoid overstating the implications of the results, while still being well-stated and linked to the original research question (Page 9, lines 374-377).

Additional comments

Dear Authors,

Thank you for submitting your manuscript titled "Perceived Masticatory Ability Evaluation in Completely Edentulous Patients with Thermoplastic Complete Denture versus Single Implant-Retained Mandibular Overdenture" to PeerJ. I have reviewed your manuscript thoroughly and would like to provide you with the following feedback.

Strengths:
1. The study compares the perceived masticatory ability of patients with thermoplastic dentures versus single implant-retained mandibular overdentures.
2. The study includes an appropriate number of participants and sample size calculation.
3. The study design, inclusion and exclusion criteria, and statistical analysis are well-described in the methodology.
4. The findings are presented clearly using tables.

Weaknesses:
1. The manuscript requires improved clarity and readability. It is recommended that a native English speaker or professional editing service proofread the manuscript.
2. The introduction should provide an overview of the current literature and highlight gaps in knowledge that the study addresses.
3. The discussion section needs a more in-depth interpretation of results and comparison with similar studies in the field.
4. The limitations of the study should be more thoroughly addressed, including the consideration of alveolar ridge height and denture retention, as these factors could influence the results.
5. The PeerJ author guidelines should be followed to properly format references, and self-citations should be minimized.

Specific comments:
1. Introduction: Provide more context on the importance of masticatory ability in completely edentulous patients and the potential benefits of thermoplastic dentures and single implant-retained mandibular overdentures. Although the current text is too lengthy, it is essential to focus on the suggested points and present the information more concisely.
2. Methods: Kindly provide an explanation for the rationale behind the selection of Polyan IC TM Bredent GmbH & Co.KG, Germany, as the thermoplastic material used in the study.
3. Results: Consider presenting the results of the Kolmogorov-Smirnov and Shapiro-Wilk tests for normality in a table format for better readability.
4. Discussion: The discussion section of your paper could be improved by providing a more detailed interpretation of the results, including a comparison of your findings with those of similar studies in the field. Additionally, it would be helpful to expand on the potential mechanisms behind the significant improvement in perceived masticatory ability with single implant-retained mandibular overdentures. This information can be found on Page 9, lines 271-289.
5. The study's limitations should be addressed more thoroughly, considering factors such as alveolar ridge height and denture retention as they could influence the results. (Page 9, lines 364-367).
6. Conclusion: Make sure that the conclusion accurately reflects the findings of the study without overstating the implications of the results.

Important: It is recommended that the manuscript be proofread by a fluent English speaker or a professional editing service to enhance its clarity and readability.

Overall, your study offers valuable insights into how completely edentulous patients perceive their masticatory ability with different prosthetic treatments. However, to enhance the manuscript's quality and increase its potential for publication in PeerJ, it would be beneficial to address the issues mentioned above.

Please consider revising your manuscript according to the feedback provided. I look forward to reviewing your revised submission.

Best regards,
Luiz Juliasse
Reviewer

---

## Round 0.2 · Minor Revisions

Please make sure to address the minor revisions suggested by the reviewers in their comments.

·

Basic reporting

In the abstract, “patent” should be replaced with “patient”. Apart from that, appropriate literature is referenced. The article is well-structured. The tables support the statements in the article. The raw data basis is shared.

Experimental design

The article is within the scope of the journal. The research question is well-defined, relevant & meaningful. It is stated how research fills an identified knowledge gap. The investigation is performed to a high technical and ethical standard. The methods are well described with sufficient detail can be replicated.

Validity of the findings

The underlying data has been provided. Conclusions are well stated, linked to original research questions and limited to supporting results.

Additional comments

Thank you for submitting the revised manuscript. The comments from the first round of reviews have been addressed.

·

Basic reporting

I used a grammar tool to improve the text slightly. All other aspects were answered by the authors.

Experimental design

The article is within the scope of the journal.

Validity of the findings

Robust data. Nice conclusions.

·

Basic reporting

1. The manuscript has undergone significant improvements in terms of language and clarity. The authors have made efforts to address most of the language-related issues mentioned in the previous review, which has enhanced the readability and understanding for an international audience. However, there are still a few areas where the English language could be further refined (please take a look at the attached PDF with comments). I recommend reviewing these sections to ensure optimal clarity.

2. The introduction section now offers a more comprehensive context regarding the significance of masticatory ability in completely edentulous patients and the potential advantages of thermoplastic dentures and single implant-retained mandibular overdentures. This enhancement effectively addresses the concerns raised in the previous review and improves the overall context of the study.

3. The authors have formatted the references in accordance with the PeerJ author guidelines and have minimized self-citations. This effort is commendable and enhances the manuscript's adherence to the journal's standards. Nonetheless, there are two highlighted issues in the attached PDF that require correction.

Experimental design

1. The study remains original primary research and falls within the scope of the journal.

2. The authors have clarified the reasoning behind the choice of the thermoplastic material used in the study, addressing the concern raised in the previous review and enhancing the understanding of the methodology.

Validity of the findings

1. The methods and results sections are well-described, and the statistical analyses are appropriate for the study design. The authors have addressed the concerns raised in the previous review, enhancing the clarity and reproducibility of the study.

2. The conclusion accurately reflects the study's findings and does not overstate the implications of the results, which is a notable improvement from the previous version.

Additional comments

I have finished reviewing the revised version of your manuscript. I want to commend you for addressing most of the concerns raised in the previous review and for the efforts you've made to improve the quality of your manuscript. I have also reviewed the supplementary tables provided and will include my assessment of these materials below.

Supplementary Tables:

1. The tables are well-organized and clearly labeled tables that support the study's findings. Also, they present relevant information, such as gender frequency, descriptive statistics of perceived masticatory ability measurements, normality test results, and comparisons between different groups and time points.

2. The tables are appropriately referenced in the main text, allowing readers to easily locate the relevant information.

3. There are explanatory footnotes for abbreviations and statistical tests used, enhancing the tables' clarity and interpretability.

4. The tables are formatted consistently and adhere to the journal's guidelines.

While the supplementary tables are informative and well-presented, I recommend providing a brief description of each table in the main text to guide the reader and highlight the main findings. This will further enhance the manuscript's readability and help readers understand the key results without having to constantly refer to the supplementary material.

General comments:

1. The discussion section has been expanded to include a more in-depth interpretation of the results and a comparison with similar studies in the field. This improvement strengthens the overall quality of the manuscript and provides valuable context for the study's findings.

2. The authors have expanded on the limitations of the study, but I recommend further discussing the potential impact of these limitations on the interpretation of the results and suggesting ways to address them in future research (Page 13, lines 346-353). This will help readers better understand the study's constraints and guide future investigations in this area.

In conclusion, the revised manuscript has significantly improved in terms of language, clarity, and depth of discussion. The authors have successfully addressed most of the concerns raised in the previous review, strengthening the quality and potential for publication of the manuscript in PeerJ. The supplementary tables are well-organized, informative, and support the study's findings. By addressing the minor issues mentioned above and providing a brief description and explanation of each supplementary table in the main text, the authors can further enhance the readability and impact of their work. I encourage the authors to consider the provided feedback and make the necessary revisions before resubmitting the manuscript for further consideration.

Best regards,
Luiz Juliasse
Reviewer

---

## Round 0.3 · accepted · Accept

Thank you for the modifications you provided.